# Risk Factor Analysis and Genetic Parameter Estimation for Pre-Weaning Mortality Traits in Boer, Spanish, and Crossbred Goat Kids

**DOI:** 10.3390/ani14071085

**Published:** 2024-04-03

**Authors:** Liuhong Chen, William Foxworth, Scott Horner, Mustafa Hitit, Negusse Kidane, Erdogan Memili

**Affiliations:** International Goat Research Center, College of Agriculture, Food and Natural Resources, Prairie View A&M University, Prairie View, TX 77446, USA; wbfoxworth@pvamu.edu (W.F.); sdhorner@pvamu.edu (S.H.); muhitit@pvamu.edu (M.H.); nfkidane@pvamu.edu (N.K.); ermemili@pvamu.edu (E.M.)

**Keywords:** pre-weaning mortality, risk factors, goat, genetic parameter

## Abstract

**Simple Summary:**

Pre-weaning mortality (PWM) is a major source of economic loss and welfare concern for goat producers. PWM is influenced by both genetic and non-genetic factors, and it varies across goat breeds and production systems. Despite its importance, there is limited research on risk factors and genetic analysis of PWM in goats. In this study, we aimed to identify fixed risk factors associated with PWM and estimate genetic parameters for PWM traits in a mixed population of purebred and crossbred Boer and Spanish goats. This study found that PWM is influenced by several factors, including birth weight, season, litter size, sex, dam age, breed, and heterosis. Furthermore, PWM has low to moderate heritability, and there are strong genetic correlations among different PWM traits defined at different pre-weaning stages. These findings provide valuable insights for goat producers to optimize their herd management and breeding programs to reduce PWM.

**Abstract:**

The objectives of this study were to evaluate fixed risk factors associated with PWM and to estimate genetic parameters for PWM. A total of 927 birth records from a mixed population of purebred and crossbred Boer and Spanish goats born between 2016 and 2023 at the International Goat Research Center (IGRC) were used for this study. Four binary traits were studied: D0–3 (death within 3 days after birth), D4–60 (death between 4 and 60 days), D61–90 (death between 61 and 90 days), and D0–90 (death within 90 days). Logistic regression models were used to evaluate the risk factors associated with PWM traits. Bayesian threshold models and Gibbs sampling were used to estimate the genetic parameters. Birth weight, season, litter size, sex, dam age, breed, and heterosis were found to be significantly associated with at least one of the PWM traits. Heritability estimates were 0.263, 0.124, 0.080, and 0.207, for D0–3, D4–60, D61–90, and D0–90, respectively. The genetic correlations between the studied traits ranged from 0.892 (D0–3 and D0–90) to 0.999 (D0–3 and D61–90). These results suggest that PWM in goats is influenced by both non-genetic and genetic factors and can be reduced by management, genetic selection, and crossbreeding approaches.

## 1. Introduction

The global goat population has more than doubled in the past four decades, reaching about one billion goats [1]. In the US, the goat population rose by over 49%, from 1.65 million in 1997 to 2.47 million in 2024 [2]. Boer and Spanish goats are two of the most widely raised meat goat breeds in the US, with different characteristics and performances. Boer goats originated from South Africa and are recognized for their high growth rate and carcass quality [3]. It has become the most popular meat goat in the US since its introduction in the 1990s. The Spanish goats are a landrace descended from the goats brought by Spanish explorers in the 1500s [4]. The Spanish goats are valued for their survivability, parasite resistance, reproduction performance, and maternal ability [5,6,7]. There has been a widespread trend to cross Boer with Spanish goats to combine the advantages of both breeds and increase the productivity and profitability of meat goat farming.

One of the major challenges that goat producers face is pre-weaning mortality (PWM), which is a significant source of economic loss and reduced productivity in goat farming. Mortality is also a farm-level indicator of animal welfare. The PWM in goats can vary between 7.8% and 46.0% [8,9] and is influenced by both non-genetic factors, such as birth weight, birth season, type of birth, sex, parity, and age of dam [8,10], and genetic factors, such as breed, individual, maternal, and heterosis effects [11,12]. Breed and heterosis effects are important to consider because they reflect the adaptation of different goat populations to various environments and production systems.

The PWM in goats can be mitigated through both management and genetic strategies. However, in the US, there is limited information on the factors affecting PWM in goats, and genetic parameters related to PWM are not reported. Browning et al. [5] found that birth month, birth type, dam breed, and dam age are significant factors affecting PWM in a Boer-Spanish-Kiko diallel crossbreeding program. Their study, however, did not differentiate between various pre-weaning stages, for which different factors may impact mortality. Literature studies have shown that most of the PWM occurred within the first few days after birth [13,14]. Estimation of genetic parameters for PWM traits is also essential for understanding the genetic basis of PWM and designing effective breeding programs to reduce it. The objectives of this study were to (1) evaluate fixed risk factors associated with PWM across different pre-weaning stages and (2) estimate genetic parameters for PWM traits in a mixed population of purebred and crossbred Boer and Spanish goats. The evaluation of these factors and parameters can provide useful insights for goat producers and breeders to reduce PWM and improve profitability.

## 2. Materials and Methods

### 2.1. Animals and Data

Purebred and crossbred goat kids from Boer and Spanish breeds were produced at the International Goat Research Center (IGRC) at Prairie View A&M University in Prairie View, Texas. The does grazed on a pasture predominantly consisting of warm-season Burmudagrass (*Cynodon dactylon*) and red clover (*Trifolium pratense*) under a semi-intensive management system. Breeding was conducted once a year from August to December in single-sire mating pens by natural services. Before mating, the does were relocated to the breeding pens next to the bucks. Synchronization was achieved using a modified 7-day CO-Synch plus a controlled internal drug release (CIDR) insert program. Sires were assigned to the does of matching breed types at random. After breeding, does returned to pastures and underwent pregnancy check 30 days later. Non-pregnant does were either culled or rebred based on their body conditions. Vaccinations against *Clostridium perfringens* types C & D and *Clostridium tetani* were administered to pregnant does 45 days before expected kidding. The does were transferred to the kidding pens 2 weeks before the due dates. Newborns received unique identification numbers, and details such as birth date, sex, birth type, and weight were documented. The kids received vaccinations for *Clostridium perfringens* types C and D and *Clostridium tetani* between 6 and 9 weeks of age. The does were moved back to the pasture after kidding, and the kids remained with their mothers until weaning. Deworming was performed for both does and kids as needed for endo-parasitism.

The crossbreeding program involved mating Boer sires with Spanish does (SB F1) and Spanish sires with Boer does (BS F1). The F1 females were then backcrossed with Boer sires (SBB) or Spanish sires (BSS). From 2016 to 2023, 927 kids were born from 233 does mated to 17 Boer and 15 Spanish sires. The breed compositions of the goat kids were: 265 Boer, 426 Spanish, 109 SB, 79 BS, and 30 BSS, and 18 SBB. High mortality was observed in two pre-weaning periods: day 0 to day 3, accounting for 55.9% of total pre-weaning loss, and day 61 to day 90, accounting for 15.3% of total pre-weaning loss. Therefore, four traits were defined for analysis: D0–3 (death within 3 days after birth), D4–60 (death between 4 and 60 days), D61–90 (death between 61 and 90 days), and D0–90 (death within 90 days). Goat kids who died before the trait start date (e.g., stillbirth) or left the herd before the trait end date were excluded from the analysis. Table 1 depicts the number and mortality of kids for each PWM trait.

The pedigree contains a total of 1116 individuals, including 51 sires and 267 dams. The pedigree was manually checked for errors, such as individuals recorded as both sire and dams or with mismatching gender information. The breed composition for crossbred individuals was deducted from the pedigree using breed information of their purebred ancestors. The pedigree structure was analyzed using the CFC software version 1.0 [15], and the details are presented in Table 2. To assess the genetic connectedness within and between the populations, Wright’s numerator relationship coefficients [16] were computed for pairs of individuals. The average genetic relationships among the populations are summarized in Table 3.

### 2.2. Risk Factor Analysis

Logistic regression models with the logit function implemented in R [17] were used to assess the effect of the risk factors on the PWM traits. The logistic regression model can be written as:$$\log\left(\frac{p}{1-p}\right)=b^{\prime }x,$$
where *p* is the probability of death, ***b*** is a vector of coefficients, and ***x*** is a vector of risk factors. The risk factors considered were: birth year, season (Winter: December–February, Spring: March–May), birth type (Single vs. Multiple), sex (Female vs. Male), birth weight, dam age, breed composition, and retained heterosis. We used the logarithm of birth year as a risk factor to test whether mortality decreased over time. Retained heterosis for each individual was calculated from the pedigree information as $RH=1-\sum_{i=1}^{n}{PS}_{i}\times {PD}_{i}$, where ${PS}_{i}$ and ${PD}_{i}$ indicate the probability that the sire and dam are from breed i, respectively. The categorical variables used in this study are binary, and therefore, odds ratios for both the continuous and categorical variables were calculated as OR=exp⁡(b) and the 95% confidence interval of the OR is calculated as CI=exp⁡(b±1.96SE) where SE is the standard error of b.

### 2.3. Genetic Parameter Estimation

We used liability-threshold models for the estimation of variance components and genetic parameters. The model assumes that the observed phenotypes are determined by an underlying continuous liability that is linked to the probability of death through a probit link function. The probability of observing a death event for animal *i* is equal to *P*($l_{i}>t),$ where $l_{i}$ is the latent liability score for animal i, and *t* is the threshold value.

A univariate model was used to estimate the heritability for each of the PWM traits, and a bi-variate model was used for estimating genetic and phenotypic correlations between pairwise PWM traits. The univariate model can be written as:l=Xb+Za+Vm+Wmpe+e,
where l is the vector of underlying liabilities; ***X***, ***Z***, ***V***, and ***W*** are the design matrix for the fixed effect, additive genetic effect, maternal genetic effect, and maternal permanent environmental effect, respectively; ***b*** is the vector of fixed effect for birth year, season, birth type, sex, birth weight, dam age, breed composition, and retained heterosis; ***a***, ***m***, ***mpe***, and ***e*** are the vectors of additive genetic effect, maternal genetic effect, maternal permanent environmental effect, and residual effect, respectively. The biological mothers of the individuals were used for both maternal genetic and maternal permanent environmental effects. The prior distribution for the additive and maternal genetic effects is $\left[\begin{array}{c} a\\ m \end{array}\right]\sim MVN\left[\begin{array}{cc} 0, & A⨂\left(\begin{array}{cc} \sigma_{a}^{2} & \sigma_{am}\\ \sigma_{am} & \sigma_{m}^{2} \end{array}\right) \end{array}\right]$, where MVN denotes multivariate normal distribution, A is the numerator relationship matrix as described by Wright [16]; $\sigma_{a}^{2}$ and $\sigma_{m}^{2}$ are the additive and maternal genetic variances, respectively, and $\sigma_{am}$ is the covariance between additive and maternal genetic effects. The prior distributions for the maternal permanent environmental and residual effects are $mpe\sim MVN(0,I\sigma_{mpe}^{2})$ and $e\sim MVN(0,\;I\sigma_{e}^{2})$, where I is the identity matrix, and $\sigma_{mpe}^{2}$ and $\sigma_{e}^{2}$ are the variances for maternal permanent environmental and residual effects, respectively. The phenotypic variance was calculated as $\sigma_{p}^{2}=\sigma_{a}^{2}+{\sigma_{am}+\sigma }_{m}^{2}+\sigma_{mpe}^{2}+\sigma_{e}^{2}.$ The direct heritability, maternal heritability, and the ratio of phenotypic variance explained by maternal permanent environmental effects were calculated by dividing the corresponding variance components by the phenotypic variance. The bivariate model can be written as:$$\left[\begin{array}{c} l_{1}\\ l_{2} \end{array}\right]=\left[\begin{array}{c} X_{1}\\ X_{2} \end{array}\right]\left[\begin{array}{c} b_{1}\\ b_{2} \end{array}\right]+\left[\begin{array}{c} Z_{1}\\ Z_{2} \end{array}\right]\left[\begin{array}{c} a_{1}\\ a_{2} \end{array}\right]+\left[\begin{array}{c} V_{1}\\ V_{2} \end{array}\right]\left[\begin{array}{c} m_{1}\\ m_{2} \end{array}\right]+\left[\begin{array}{c} W_{1}\\ W_{2} \end{array}\right]\left[\begin{array}{c} {mpe}_{1}\\ {mpe}_{2} \end{array}\right]+\left[\begin{array}{c} e_{1}\\ e_{2} \end{array}\right],$$
where all the model items are as defined previously but with subscripts 1 and 2 denoting trait 1 and trait 2, respectively. The prior distribution for the additive and maternal genetic effects are: $\left[\begin{array}{c} a_{1}\\ a_{2} \end{array}\right]\sim MVN\left[\begin{array}{cc} 0, & A⊗\left(\begin{array}{cc} \sigma_{a_{1}}^{2} & \sigma_{a_{1}a_{2}}\\ \sigma_{a_{1}a_{2}} & \sigma_{a_{2}}^{2} \end{array}\right) \end{array}\right]$, and $\left[\begin{array}{c} m_{1}\\ m_{2} \end{array}\right]\sim MVN\left[\begin{array}{cc} 0, & A⊗\left(\begin{array}{cc} \sigma_{m_{1}}^{2} & \sigma_{m_{1}m_{2}}\\ \sigma_{m_{1}m_{2}} & \sigma_{m_{2}}^{2} \end{array}\right) \end{array}\right]$, where $\sigma_{a_{1}a_{2}}$ and $\sigma_{m_{1}m_{2}}$ are the additive genetic and maternal genetic covariances between traits 1 and 2, respectively. The prior distribution for the maternal permanent environmental and residual effects are $\left[\begin{array}{c} {mpe}_{1}\\ {mpe}_{2} \end{array}\right]\sim MVN\left[\begin{array}{cc} 0, & I⊗\left(\begin{array}{cc} \sigma_{{mpe}_{1}}^{2} & \sigma_{m{pe}_{1}{mpe}_{2}}\\ \sigma_{{mpe}_{1}{mpe}_{2}} & \sigma_{m{pe}_{2}}^{2} \end{array}\right) \end{array}\right],$ and $\left[\begin{array}{c} e_{1}\\ e_{2} \end{array}\right]\sim MVN\left[\begin{array}{cc} 0, & I⊗\left(\begin{array}{cc} \sigma_{e_{1}}^{2} & \sigma_{e_{1}e_{2}}\\ \sigma_{e_{1}e_{2}} & \sigma_{e_{2}}^{2} \end{array}\right) \end{array}\right]$, where $\sigma_{m{pe}_{1}{mpe}_{2}}$ and $\sigma_{e_{1}e_{2}}$ are the covariances for maternal permanent environmental and residual effects between traits 1 and 2, respectively.

The parameters were estimated using Gibbs sampling algorithms implemented in the THRGIBBS1F90 program [18]. The Gibbs sampling process was run for 500,000 cycles, with a burn-in period of 75,000 cycles. The convergence and the burn-in period of the chain were determined by the Geweke convergence diagnostic statistic [19]. The remaining chain was thinned by keeping every 50th sample. Posterior means and standard deviations of the samples were computed using the POSTGIBBSF90 program [18]. The phenotypic covariance between traits 1 and 2 was calculated as $\sigma_{p_{1}p_{2}}=\sigma_{a_{1}a_{2}}+\sigma_{m_{1}m_{2}}+\sigma_{m{pe}_{1}{mpe}_{2}}+\sigma_{e_{1}e_{2}}$, and the genetic and phenotypic correlations were calculated as $\sigma_{a_{1}a_{2}}/\sqrt{\sigma_{a_{1}}^{2}\times \sigma_{a_{2}}^{2}}$ and $\sigma_{p_{1}p_{2}}/\sqrt{\sigma_{p_{1}}^{2}\times \sigma_{p_{2}}^{2}}$, respectively.

## 3. Results

### 3.1. Logistic Regression Analysis of Risk Factors

The logistic regression analysis revealed several significant risk factors associated with the four PWM traits in the current evaluated goat population. Table 4 presents the estimates of the effects, OR and their corresponding 95% CI, and *p*-values for the risk factors. Birth season was significantly (*p* < 0.05) associated with D0–3, with an estimated effect of 0.60 and an OR of 1.82. This indicates higher mortality during winter compared to spring. Birth weight showed a highly significant (*p* < 0.001) effect on D0–3, D4–60, and D0–90, and a significant (*p* = 0.008) effect on D61–90, with estimates ranging from −0.33 to −0.44 and ORs ranging from 0.65 to 0.72, suggesting that increased birth weight is linked to reduced mortality risk. Birth type also had a significant (*p* < 0.05) effect on D0–3, with an estimate of −1.42 and an OR of 0.24, indicating higher mortality in multiple births than in single births. Breed had a significant (*p* < 0.05) effect on D4–60 and a highly significant (*p* < 0.001) effect on other traits, with estimates ranging from −0.76 to −2.61 and ORs ranging from 0.07 to 0.47, suggesting reduced mortality in the Spanish breed compared to Boer breed. Retained heterosis was also significant (*p* = 0.009) for D0–3 and highly significant (*p* < 0.001) for D0–90, with estimates of −0.86 and −0.93 and ORs of 0.42 and 0.39, respectively, suggesting that heterosis contributed to reduced mortality. Dam age had a significant (*p* < 0.05) effect on D4–60 and D0–90, with estimates of 0.14 and 0.09 and ORs of 1.15 and 1.10, respectively, implying that older dams had higher mortality rates of their offspring. Sex had a highly significant (*p* < 0.001) effect on D0–90, with an estimate of −0.69 and an OR of 0.50, indicating lower mortality in females than males. Birth year has no significant (*p* > 0.05) effect on any of the traits, indicating no temporal trend in mortality in the studied population.

### 3.2. Estimates of Genetic Parameters

The posterior means and standard deviations of the variance components and the heritability estimates for the four PWM traits are presented in Table 5. The highest direct heritability was observed for D0–3 (0.263), followed by D0–90 (0.207), D4–60 (0.124), and D61–90 (0.080). The maternal heritability for the PWM traits ranged from 0.055 (D4–60) to 0.121 (D0–3), and the proportion of total variance explained by maternal permanent environmental effects varied from 0.027 (D61–90) to 0.043 (D0–3). The results indicate that there is substantial genetic variation for PWM in goats among individuals and that genetic selection can improve the survival of the kids. The results also suggest that better maternal ability and care of the does can enhance the survivability of the kids.

The posterior means and standard deviations of the additive genetic correlations and phenotypic correlations among the four PWM traits are depicted in Table 6. The genetic correlations were very high for all pairs of traits, ranging from 0.892 (D0–3 and D0–90) to 0.999 (D0–3 and D61–90). These results indicate that there are genes that affect multiple PWM traits, and selection for one trait would have a strong effect on the others. However, the genetic correlations were not perfect, implying that some genes were specific to certain PWM traits. The phenotypic correlations were also high, ranging from 0.666 (D0–3 and D0–90) to 0.998 (D61–90 and D0–90). The lower phenotypic correlation between D0–3 and D0–90 and D4–60 and D0–90 indicate that environmental factors affect different PWM traits at different periods.

## 4. Discussion

### 4.1. Risk Factors Associated with Pre-Weaning Mortality

Results from our logistic regression analysis showed that birth weight is consistently associated with PWM across all the traits studied. This finding is consistent with previous studies that reported a negative relationship between birth weight and mortality during the pre-weaning period [8,20,21]. According to [22], a 1 kg increase in kid birth weight reduces the probability of mortality by 32.5%. Luo et al. [9] also found that when the birth weight was 20% lower than the annual average, the PWM increased to 46%. Hailu et al. [23] also reported a maximum survival rate of 74% from kids greater than 3 kg at birth. These results suggest that birth weight is a crucial factor in the survival of goat kids. Heavier kids are likely to have more sufficient energy reserves to maintain body heat and stronger immunity to resist diseases and environmental hazards [24,25]. Therefore, appropriate management, including feeding with colostrum, milk replacers, and brooding with artificial heating for kids with light weights, is needed to improve their survival.

Birth type had a significant effect on trait D0–3 but not on the other traits from day 4 to day 90, indicating that multi-born kids are more vulnerable in the early postnatal period than single-born kids. One likely reason for this is that multi-born kids tend to have lower birth weights than single-born kids, which may impair their ability to cope with environmental hazards or nurse colostrum from their mothers shortly after birth. In this study, the average birth weight was 3.65 kg for single-born kids and 3.28 kg for multi-born kids. Moreover, multi-born kids may face competition from their siblings for accessing their mothers’ teats, which may further reduce their colostrum intake. Previous studies have also reported higher mortality rates for multi-born kids than single-born kids in various breeds [8,26,27,28,29]. This study did not separate the effects of twin-born and triplet-born effects due to the relatively low number of records. However, literature studies have reported that triplet-born kids had higher mortality than twin-born kids [27,28,29]. Therefore, it is recommended that multi-born kids, especially those with low birth weights, receive special care and management to improve their survival and growth performance.

Birth season had a significant influence on trait D0–3 and a marginal influence on trait D61–90. However, the direction of the effect was the opposite for these two traits. Winter-born kids had higher mortality for D0–3 and lower mortality for D61–90, while spring-born kids had lower mortality for D0–3 and higher mortality for D61–90. The higher mortality rate in D0–3 for winter-born kids may be attributed to the cold weather in winter. According to historical weather data, the average temperature between December and February 2016 and 2023 in Texas is 10.8 °C (51.4 °F). The higher mortality rate in D61–90 for spring-born kids may be related to the hotter temperature in summer around the weaning period. The average temperature between May and August in Texas is 28.3 °C (82.9 °F). Previous literature studies have reported that both cold and hot weather can affect PWM. Luo et al. [9] found that the mortality for the first month after birth increases when temperature is below 11 °C (52 °F), while the mortality between 2 and 4 months after birth increases when temperature is greater than 26 °C (79 °F). Niverthikaa et al. [30] identified cold weather and heat stress as two of the major factors affecting PWM in non-native goat breeds reared in the Ampara District in Sri Lanka. Snyman [27] from South Africa also found that season had a significant effect on kid mortality in Angora goats, with higher mortality rates in winter and summer than in autumn and spring. These results suggest that seasonal variations affect kid mortalities significantly, and management practices should be adjusted accordingly to reduce the risks. Hailu et al. [23] also suggested that during the spring season, does could have access to sufficient vegetation to produce milk for their young offspring compared to other seasons of the year.

Male kids had higher PWM than female kids, which is consistent with previous studies in different goat breeds [8,23,26,31]. The sex effect was more noticeable in the later stages of pre-weaning than in the early stages. The reason for low survivability in male goat kids is unclear, but it may be associated with reduced thermoregulation and behaviors such as standing, udder seeking, and sucking ability [8]. Male-linked mortality has also been reported in other species, such as piglets [32], sheep [33,34], and humans [35]. Further research is required to determine the causal factors affecting male kid PWM.

The PWM rate increased with the age of the dam in general. The higher mortality in older dams could be due to the decline in maternal ability, milk production, and immune function [10,28,36]. Our data showed that the mortality rates for kids born from dams aged 1, 2, 3, 4, 5, and 6+ years were 28.6%, 10.7%, 23.5%, 29.9%, 29.7%, and 30.2%, respectively. The lowest mortality rate was observed in kids born from dams aged 2 years, which might be the optimal age for reproduction and lactation in goats. Kids born from dams aged 1 year had a higher mortality rate than those born from dams aged 2 and 3 years. A possible reason is that these 1-year-old does were bred before reaching their optimal reproductive performance. It is worth noting that parity information was not recorded in our dataset. Other studies evaluated the effect of parity instead of dam age and consistently found that first-parity does tend to have higher mortality in their offspring [8,26,29]. First-parity does may have low milk production and colostrum quality. They may also have less maternal experience and ability to care for their kids, especially if they have multiple births.

The Boer breed had higher mortality rates across all the pre-weaning traits studied compared to the native Spanish breed. This is consistent with previous studies that showed the Boer breed had poor survival rates compared to other breeds in different countries and regions. For example, Tesema et al. [37] and Tessama et al. [38] reported that the Boer breed had higher mortality rates than the local Ethiopian breeds. Harrison [39] showed that Boer and its crosses with Kiko had higher mortality rates compared to purebred Kiko breed in the US. These results suggest that the Boer breed is less adapted to the environment and management system of the study areas despite its higher growth rate. Crossbreeding can improve pre-weaning survival chances by introducing hybrid vigor. However, the effect of crossbreeding may vary depending on the breeds and environments involved. Pérez-Baena et al. [40] observed that Boer-crossed kids had significantly lower mortality than Murciano-Granadina purebred kids in Spain. Harrison [39] showed the crossbred Boer and Kiko had lower mortality than the purebred Boer but still significantly higher mortality than the purebred Kiko. In this study, the heterosis effect reduced the probability of PWM by up to 61%, suggesting the potential of crossbreeding Boer with the local Spanish breed in the US.

### 4.2. Genetic Parameters for Pre-Weaning Mortality Traits

Genetic parameter estimates for PWM traits in goats are limited in the literature. Previous studies have reported low heritability estimates for PWM, ranging from 0.03 to 0.10 in different goat breeds and populations. For example, Synman [27] estimated a heritability of 0.04 for PWM in South African Angora goats. Rout et al. [41] estimated heritability of 0.03 in Jamunapari goats in India. Josiane et al. [42] estimated a heritability of 0.04 in indigenous goats in Burundi. Tesema et al. [43] estimated heritability of 0.10 in a crossbred population of Boer and Central Highland goats in Ethiopia. Our estimates of heritability were higher than those reported in the literature. This could be due to the genetic diversity of the population we studied, which consisted of Boer, Spanish, and their crossbreds. Another possible explanation is that most of the previous studies used field data, which may have more measurement errors and unaccounted environmental factors. However, the number of records used in our study was still relatively small, which resulted in large standard deviations of the estimates. Therefore, more data are needed to obtain more accurate estimates of the genetic parameters for PWM in our population.

Heritability estimates for different periods of PWM in goats are lacking in the literature. Several studies conducted in sheep have reported low heritability estimates in different pre-weaning periods. Sallam [44] reported that heritability estimates ranged from 0.011 to 0.020 for D0–3, D4–60, D61–90, and D0–90 in Barki lamb in Egypt. Brien et al. [45] found similar results for D0–3 and D4–85 in different sheep breeds in Australia, with heritability estimates of 0.014 and 0.010, respectively. Vostry and Milerski [46] reported heritability estimates for D0–1 (0.024) and D2–14 (0.033) in different sheep breeds in the Czech Republic. Our results suggest that the direct heritability was highest for D0–3 (0.263) and lowest for D61–90 (0.080), which is reasonable as more random environmental effects will likely affect mortality in later stages. The results suggest that selecting for D0–3 could be a viable strategy for improving PWM. One advantage of this strategy is that D0–3 has the highest heritability, and therefore, more genetic gain can be achieved than selection for other traits. Furthermore, more phenotypic records can be used for D0–3 as some records will be missing for traits that are recorded later as animals leave the herds for varied reasons, such as slaughter or sale.

Our results showed that maternal genetic effects explained less variation than additive genetics but more than maternal permanent environmental effects. Previous studies have reported different maternal effects for PWM in goats. For example, Rout et al. [41] estimated a maternal permanent environmental effect of 0.01 in Jamunapari goats. Josiane et al. [42] estimated a maternal genetic heritability of 0.04, similar to the direct heritability, in indigenous goats in Burundi. Tesema et al. [43] estimated a maternal permanent environmental effect of 0.06 in Boer-Central Highland crossbred goats. The combined maternal effects in our study, both genetic and environmental, accounted for around 8–16% of the phenotypic variance. This suggests that the does’ selection and management are important for their kids’ survival.

Genetic correlations among different time periods of PWM for goats have not been reported in the literature. However, several studies have estimated these correlations for other livestock species. For example, Hansen et al. [47] estimated a genetic correlation of 0.73 between D1–14 and D15–60 in Danish Holstein cattle. Sallam [44] reported moderate to high genetic correlations among PWM traits, ranging from 0.72 (D0–3 and D4–60) to 0.95 (D4–60 and D61–90) in Barki lamb in Egypt. Ibi et al. [48] estimated a moderately high genetic correlation of 0.69 between D15–60 and D61–180, 0.50 between D1–14 and D15–60, but low genetic correlation of 0.06 between D1–14 and D61–180 in Japanese Black beef cattle. The positively high genetic correlations between pairs of the traits in our study suggest that mortality at different pre-weaning periods may share some common genes or genetic factors. However, these correlations may vary depending on the population and environment. Therefore, more studies are needed to estimate the genetic correlations among PWM in different time periods for different goat populations.

## 5. Conclusions

Pre-weaning mortality in goats is affected by both genetic and non-genetic factors. Our study identified birth season, birth type, sex, birth weight, dam age, breed, and heterosis as significant risk factors for this trait. There are substantial genetic variations for PWM among individuals, as well as maternal effects. To reduce PWM in goats, it is essential to account for all the factors that influence this economically important trait. Better management practices and genetic approaches, such as selective breeding and crossbreeding that exploit breed and heterosis effects, can be used to improve the pre-weaning survival of goat kids. The reason for lower survivability in male goat kids compared to female goat kids is unclear, necessitating further research to pinpoint the primary cause of the observed higher mortality rate in male goat kids.

## Figures and Tables

**Table 1 animals-14-01085-t001:** Number and mortality of goat kids for the pre-weaning mortality traits ^1^.

Trait	Kids Alive (N)	Kids Died (N)	Total Kids (N)	Mortality (%)
D0–3	828	99	927	10.68
D4–60	691	51	742	6.87
D61–90	556	27	583	4.63
D0–90	556	177	733	24.15

^1^ D0–3: death within 3 days after birth; D4–60: death between 4 and 60 days after birth; D61–90: death between 61 and 90 days after birth; D0–90: death within 90 days after birth.

**Table 2 animals-14-01085-t002:** Pedigree structure of Boer, Spanish, crossbred populations.

Item	Boer	Spanish	Crossbred ^1^	Overall
Number of individuals in the pedigree	388	490	414	1116
Number of sires in the pedigree	32	15	38	51
Number of dams in the pedigree	101	120	156	267
Number of animals with both parents	261	422	331	920
Number of animals with one parent	28	11	4	39
Number of animals with no parents	99	57	79	157
Average pedigree depth	2.117	2.178	2.597	2.36
Maximum pedigree depth	5	5	5	5
Average inbreeding coefficient	0.004	0.013	0.001	0.007
Maximum inbreeding coefficient	0.25	0.25	0.25	0.25

^1^ The pedigree for the crossbred population includes crossbred individuals and their purebred ancestors.

**Table 3 animals-14-01085-t003:** Average genetic relationships within and between the Boer, Spanish, and Crossbred populations.

Population	Boer	Spanish	Crossbred
Boer	0.029	0.000	0.022
Spanish		0.061	0.029
Crossbred			0.058

**Table 4 animals-14-01085-t004:** Estimates of the effect (b ± SE), odds ratio (OR), 95% confidence interval (CI), and *p*-values for the risk factors associated with the pre-weaning mortality traits ^1^.

Trait	Risk Factor	b ± SE	OR (95% CI)	*p*-Value
D0–3	Birth year	0.14 ± 0.22	1.15 (0.75, 1.80)	0.535
	Season (Winter)	0.60 ± 0.24	1.82 (1.13, 2.93)	0.013
	Sex (Female)	−0.23 ± 0.23	0.79 (0.51, 1.24)	0.306
	Birth weight	−0.33 ± 0.08	0.72 (0.61, 0.85)	<0.001
	Birth type (Single)	−1.42 ± 0.61	0.24 (0.06, 0.68)	0.019
	Dam age	0.06 ± 0.05	1.06 (0.96, 1.17)	0.253
	Breed (Spanish)	−1.53 ± 0.27	0.22 (0.13,0.37)	<0.001
	Heterosis	−0.86 ± 0.33	0.42 (0.22,0.79)	0.009
D4–60	Birth year	−0.05 ± 0.27	0.96 (0.57, 1.67)	0.869
	Season (Winter)	0.05 ± 0.34	1.05 (0.54, 2.02)	0.874
	Sex (Female)	−0.52 ± 0.31	0.59 (0.32, 1.09)	0.094
	Birth weight	−0.39 ± 0.12	0.68 (0.54, 0.86)	0.001
	Birth type (Single)	−0.39 ± 0.55	0.68 (0.20, 1.77)	0.476
	Dam age	0.14 ± 0.06	1.15 (1.02, 1.30)	0.019
	Breed (Spanish)	−0.76 ± 0.36	0.47 (0.23, 0.96)	0.035
	Heterosis	−0.45 ± 0.43	0.64 (0.26, 1.44)	0.297
D61–90	Birth year	0.15 ± 0.43	1.17 (0.52, 2.90)	0.724
	Season (Winter)	−1.12 ± 0.58	0.33 (0.09, 0.94)	0.055
	Sex (Female)	−0.82 ± 0.46	0.44 (0.18, 1.07)	0.072
	Birth weight	−0.39 ± 0.15	0.67 (0.50, 0.90)	0.008
	Birth type (Single)	−0.01 ± 0.67	0.99 (0.22, 3.26)	0.990
	Dam age	0.00 ± 0.11	1.00 (0.79, 1.24)	0.984
	Breed (Spanish)	−2.61 ± 0.57	0.07 (0.02, 0.21)	<0.001
	Heterosis	−0.77 ± 0.69	0.46 (0.10, 1.64)	0.261
D0–90	Birth year	0.00 ± 0.19	1.00 (0.69, 1.46)	0.993
	Season (Winter)	0.08 ± 0.21	1.09 (0.72, 1.64)	0.686
	Sex (Female)	−0.69 ± 0.20	0.50 (0.34, 0.74)	<0.001
	Birth weight	−0.44 ± 0.08	0.65 (0.56, 0.75)	<0.001
	Birth type (Single)	−0.70 ± 0.37	0.50 (0.23, 0.99)	0.060
	Dam age	0.09 ± 0.04	1.10 (1.01, 1.19)	0.030
	Breed (Spanish)	−1.67 ± 0.24	0.19 (0.12, 0.30)	<0.001
	Heterosis	−0.93 ± 0.27	0.39 (0.23, 0.66)	<0.001

^1^ D0–3: death within 3 days after birth; D4–60: death between 4 and 60 days after birth; D61–90: death between 61 and 90 days after birth; D0–90: death within 90 days after birth.

**Table 5 animals-14-01085-t005:** Posterior estimates of means (and standard deviations) for variance components and heritability values for the pre-weaning mortality traits ^1,2^.

Trait	$\sigma_{a}^{2}$	$\sigma_{m}^{2}$	$\sigma_{mpe}^{2}$	$\sigma_{e}^{2}$	$h_{d}^{2}$	$h_{m}^{2}$	$c^{2}$
D0–3	0.028 (0.023)	0.012 (0.006)	0.004 (0.004)	0.058 (0.013)	0.263 (0.190)	0.121 (0.059)	0.043 (0.037)
D4–60	0.009 (0.007)	0.004 (0.002)	0.002 (0.001)	0.053 (0.005)	0.124 (0.098)	0.055 (0.030)	0.027 (0.021)
D61–90	0.004 (0.003)	0.003 (0.002)	0.001 (0.001)	0.036 (0.003)	0.080 (0.071)	0.073 (0.036)	0.027 (0.024)
D0–90	0.038 (0.024)	0.012 (0.007)	0.006 (0.005)	0.121 (0.016)	0.207 (0.119)	0.068 (0.038)	0.034 (0.029)

^1^ $\sigma_{a}^{2}$: additive genetic variance; $\sigma_{m}^{2}$: maternal genetic variance; $\sigma_{mpe}^{2}$: maternal permanent environmental variance; $\sigma_{e}^{2}$: residual variance; $h_{d}^{2}$: direct heritability; $h_{m}^{2}$: maternal genetic heritability; $c^{2}$: proportion of phenotypic variance explained by maternal environmental effects. ^2^ D0–3: death within 3 days after birth; D4–60: death between 4 and 60 days after birth; D61–90: death between 61 and 90 days after birth; D0–90: death within 90 days after birth.

**Table 6 animals-14-01085-t006:** Posterior estimates of means (and standard deviations) for genetic correlations (above diagonal) and phenotypic correlations (below diagonal) among the pre-weaning mortality traits ^1^.

Trait	D0–3	D4–60	D61–90	D0–90
D0–3		0.998 (0.003)	0.999 (0.002)	0.892 (0.123)
D4–60	0.996 (0.004)		0.998 (0.004)	0.941 (0.081)
D61–90	0.996 (0.002)	0.996 (0.003)		0.997 (0.010)
D0–90	0.666 (0.029)	0.843 (0.020)	0.998 (0.005)	

^1^ D0–3: death within 3 days after birth; D4–60: death between 4 and 60 days after birth; D61–90: death between 61 and 90 days after birth; D0–90: death within 90 days after birth.

## Data Availability

The data presented in this study are available from the corresponding authors upon reasonable request. The data is not publicly available due to privacy or ethical restrictions.

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
