# Peer review of "Risk Factor Analysis and Genetic Parameter Estimation for Pre-Weaning Mortality Traits in Boer, Spanish, and Crossbred Goat Kids"

_animals, 2024, doi:10.3390/ani14071085_

Round 1

Reviewer 1 Report

Comments and Suggestions for Authors

This paper describes an analysis of the risk factors and genetic parameters associated with PWM in goats. The experimental design is generally simple and straightforward, and the authors largely achieve their objectives. However, the main concern for this reviewer is the need for clarification regarding the genetic terminology and statistical methods employed.

line 109: OR=exp(b) only works for continuous predictors in logistic regression model. Please add the description for calculating odds ratio for categorical predictors.

line 114: What is liability, and how is it calcuated?  

line 120: What does "genetic correlations between pairwise PWM traits" mean? The genotypes are expected to remain consistent across different time points. Does it refer to different sets of trait-associated SNPs?

line 122: If the liability represents the probability of death, its value falls within the range of 0 to 1. However, for the dependent variable, it is expected to span from negative infinity to positive infinity, unless a specific link function is employed. Please clarify.

line 124: How were the maternal genetic effect and maternal permanent environmental effect encoded?

line 128: MVN --multivariate normal distribution, please provide the full name.

line 129: Additive refers to the genetic effect and is coded by genotype. What does it mean by "additive genetic relationship matrix calculated from the pedigree"?

line 199: What does "polymorphic genes" mean?

Reviewer 2 Report

Comments and Suggestions for Authors

Thank you for your efforts in conducting the risk factor analysis and genetic parameter estimation for pre-weaning mortality traits in Boer, Spanish, and Crossbred goat kids. The findings hold significant potential for the scientific community. However, there are certain areas that should be addressed to enhance the manuscript's potential for publication.

1. The background of the study appears to be somewhat lacking in depth. I recommend strengthening the background section to enhance the context and relevance of the research.

2. The population structure of Boer, Spanish, and Crossbred goats is important.

3. Please describe the management conditions of the studied populations more elaborately.

4. Provide pedigree structure information for the studied population, and if possible, include it in a table.

5. Did the author check the inbreeding coefficient and average relatedness of the studied population? If not, why not? Please explain.

6. Clearer explanations of the fixed effects used in genetic parameter estimation are needed.

Addressing these points will likely enhance the manuscript's quality and increase its chances of acceptance.

Round 2

Reviewer 1 Report

Comments and Suggestions for Authors

The authors have addressed my comments. I am happy with the content of this manuscript.